# Seeing the Forest for the Trees: A Review-Based Framework for Better Harmonization of Timber Production, Biodiversity, and Recreation in Boreal Urban Forests

**Vegard Gundersen \*, Berit Köhler and Knut Marius Myrvold**

Lillehammer, Norsk institutt for naturforskning, Vormstuguvegen 40, 2624 Lillehammer, Norway;
Berit.Kohler@nina.no (B.K.); Knut.Myrvold@nina.no (K.M.M.)

\* Correspondence: vegard.gundersen@nina.no

**Abstract:** Forested lands serve multiple needs, and the priorities that go into balancing the competing demands can vary over time. In addition to being the source of timber and other natural resources, forested lands provide a number of other services such as biodiversity conservation and opportunities for outdoor recreation. While allocations that enhance conservation and recreation can involve expenses and lost revenue, mechanisms exist to provide landowners with incentives to make such contributions. Here, we review the literature and present a conceptual framework that can help landowners envision possible contributions towards bolstering outdoor recreation opportunities on their lands. The framework classifies forests within a simple conceptual space defined by two axes: (1) the spectrum of intensity of recreational use, and (2) the level of economic contribution required by landowners to meet recreational demands of visitors to their lands. The resulting matrix consists of four broad categories that can be used in forest management zoning as seen from an outdoor recreation perspective: general and special considerations for recreational opportunities and biodiversity, wilderness and nature reserves, and service areas. These categories have different tolerances for active silviculture and require shifting harvest practices spatially within the forest property. While timber revenues may decrease with shifting allocations, other sources of revenue may open up. With an increasingly urban population and rising demands for natural resources, it is prudent for landowners and land use planners to consider zoning their properties to better handle potential conflicts. The framework presented here provides a simple, structured approach to visualize future challenges and opportunities.

**Keywords:** biodiversity; Fennoscandia; leisure time; recreation; visitor facilities; urban forests

---

## 1. Introduction

Urban land management decisions for both publicly and privately owned forest involve more than simply maximizing natural resource production [1]. In addition to being the source of timber and other resources, forested lands provide a number of other services such as wildlife habitat, carbon sequestration, and opportunities for outdoor recreation. Despite the fast growing body of research literature within market and non-market ecosystem services, trade-offs between services are still poorly understood [2]. Outdoor recreation is important on all forest land and is one of the most highly valued non-market services for all urban ecosystems [2–7]. The links between biodiversity, timber production, and recreation are well-studied [8–13], but studies rarely result in practical recommendations for how managers could integrate these values into the management of urban forest landscapes [14].

In recent decades, overarching initiatives like the Convention on Biological Diversity (CBD) have slowly started to influence national policies for forest management. Furthermore, heightened ecological awareness among public land managers and private landowners has led to policies that also consider preservation of biodiversity for all managed landscapes [15]. The combination of legal restrictions and market-based incentives established to preserve biodiversity by preserving habitat in natural and semi-natural landscapes often creates areas that are also attractive for visitors [16,17]. Therefore, outdoor recreation opportunities should be regarded as a product or service provided by forested ecosystems in much the same way as these ecosystems produce raw materials such as timber and preserve biodiversity [5,18]. Whereas land management policies that favor biodiversity can also enhance opportunities for recreation [12], there is often little spatial overlap between areas valuable to biodiversity conservation and frequently visited recreation areas [17,19,20]. In planning for both biodiversity conservation and recreational opportunities, it becomes important to identify where different activities can take place and take steps to secure access to these areas, much in the same way we preserve the most biodiverse portions of the forest.

In general, people's aesthetic preferences for recreation align with areas that are valuable for the protection of biodiversity. Except in areas with heavy use, conflicts between the two were found to be rare in urban woodlands in recent studies [12,21]. A managed, semi-open, park-like forest with low biodiversity may have conventional scenic appeal [22,23], but people who are familiar with the ecological importance of old trees and dead wood might find natural forests more attractive [24]. Understanding the relationships between preferences for outdoor recreation and areas that are important to biodiversity conservation is hence important to avoid conflict. There are several examples where people's preferences for scenic landscape do not align with the dynamics of natural forest, e.g., fire and post-fire succession, large-scale insect attacks, downfall, complex successional structures, and dead and dying trees [24–26]. This provides both opportunities and limitations in the spatial arrangement of zones intended to preserve one or the other set of values. In addition to considering biodiversity, managers should also consider access to a range of outdoor recreational activities that may or may not overlap with the areas set aside for biodiversity conservation [8,24,27].

Balancing the competing demands of timber production, biodiversity conservation, and access to outdoor recreation inherently requires a spatial zoning approach, and is inevitably context-dependent: Timber production can degrade biodiversity and limit the attractiveness of recreation; outdoor recreation can jeopardize conservation values if volumes are above the area's tolerance; and strict conservation regimes can exclude timber production and recreation opportunities [1,3]. To help illustrate these dilemmas, we propose a simple conceptual framework for characterizing the recreational suitability of forests and woodlands when balancing resource extraction, biodiversity conservation, and recreational opportunities.

We review the literature pertaining to outdoor recreational opportunities with regards to forest management incentives and people's preferences for forest structures. We then use the recreational opportunity spectrum (ROS) to conceptually identify different zones of forested lands in which forest production, biodiversity conservation, and outdoor recreation are emphasized. Finally, we provide a discussion of the framework's relevance to European forest management. Our goal is to illustrate how forested lands can offer a range of recreational opportunities with varying degrees of development in addition to producing timber and providing habitat for biodiversity.

## 2. Methodology and Delimitation of the Review

### 2.1. Delimitation of the Review

We chose to focus on Fennoscandian boreal systems; however, studies from other parts of Europe were included if they contributed toward understanding recreational in boreal systems or discussed knowledge transfer between European forest systems (e.g., broadleaved evergreen, thermophilic deciduous). The Fennoscandian boreal forest landscape differs from the European continent with

regard to management regimes, site conditions, history of use, and socio-economic value [2]. First, in Fennoscandia, the conifer forest is the primary landscape element and is important for the economy and for national identities. Forestry has been an important industry in Fennoscandia and the practices there have been amongst the most efficient and mechanized in the world for several decades. Most of the boreal forested land is natural or semi-natural land that historically has been formed by large-scale disturbance (e.g., fire and windthrows), and that today includes plantations or self-regenerating natural tree species. In addition, the landscapes surrounding urban areas tend to be largely forested with a sharp urban-forest gradient [1]. Next, there is public access to all forest land in Fennoscandia, which is important for people's continued connections to forested landscapes in the face of increasing urbanization. People use the forest intensively for harvesting (e.g., mushroom and berry picking, which is permitted for everyone) and recreation (e.g., skiing and hiking). These northern traditions are formed by—and form—the dominance of boreal or boreonemoral forests on the landscape. We contend that the position of forests in this society differs from other parts of Europe. For example, landscapes in central Europe are more patchily forested, include large proportion of hardwoods, and have a long history of private management and use, whereas the western parts of Europe and the United Kingdom have very low percentages of woodlands remaining, of which most consist of plantations and exotic species [1]. These factors make it difficult to compare Fennoscandian boreal conifer forests to other types of forests in the rest of Europe, which is an important reason why managers of European boreal forests often looked to North America for theoretical concepts and practical management [17].

## 2.2. Reviewed Literature

Reviews of the interface between timber production, recreation, and conservation have previously been published [8–13]. Consequently, we did not perform a comprehensive review in this paper, but merely focused on influential reviews and seminal papers as a starting point for a targeted search in a restricted geographical area. However, for the quite narrow topic of people's forest preferences, we did a comprehensive search with the aim to include all relevant scientific papers from Fennoscandia. For this we used four international databases (Web of Science, Google Scholar, Oria, and Scopus) with a diversity of terms: preference, perception, attitude, like, dislike, visual, scenic, appreciation, aesthetic, and expectation with the word boreal, forest, wood, park, and woodland in combination with Norway, Sweden, Finland, and other Nordic regions. We included all peer-reviewed papers that addressed people's visual preferences for forest since the topic first appeared in 1972. We evaluated 152 papers, resulting in 104 papers for this review (Figure 1). Some of the papers included two or more surveys; some papers were based on the same dataset; and some of the papers focused on other landscape components (urban parks, rivers, agricultural lands) but included important results about visual forest attributes. The 48 papers that were excluded from the review provided important baseline information about mechanisms for forest visual appreciation, but did not directly investigate visual preferences. We focused on the boreal forest in Fennoscandia (Finland, Norway, and Sweden) as integrating these values has been an important issue over the last fifty years [17]. Clear-cutting and the construction of forest roads have caused large conflicts between forestry and recreationists, and there was a need for knowledge about people's preferences to solve these practical problems in forestry [28]. The survey methodology has varied substantially, using different kinds of target populations (national, regional, and local), sampling modes (on-site, mailed, web-based), and stimuli (photos, verbal, on-site in the forest environment).

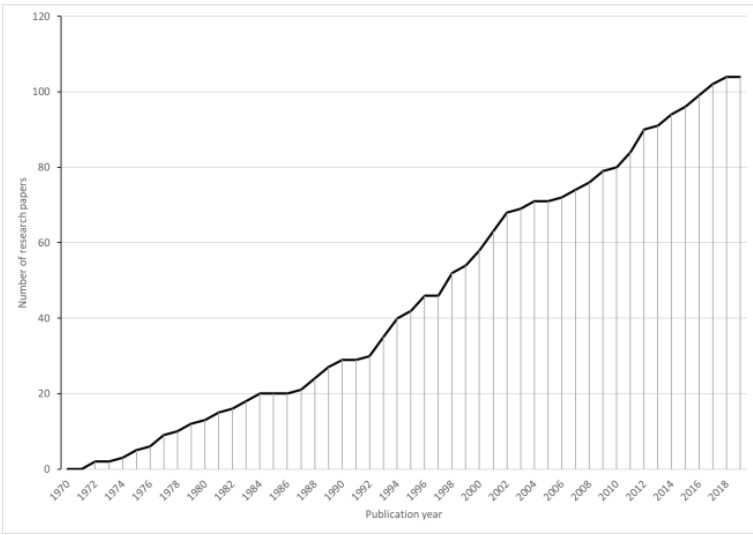

**Figure 1.** Cumulative curve for published Fennoscandian quantitative research papers including forest preferences surveys in the period from 1972 to 2019.

The results from these surveys are derived from the most common answers among respondents (median values), or from their average ratings (mean values). No survey design can prevent biases such as overrepresentation by certain groups, such as active outdoor recreationists who have an interest in forest nature and people who feel comfortable with answering multiple-choice questions or providing ratings. In interpreting the results, the reader should keep in mind the diversity of responses, the motivations of people who did not respond, or who were not even invited to respond. Furthermore, children, young adolescents, old people, and immigrants (i.e., those that last generation have migrated to the country) were strongly underrepresented in the surveys. Quantitative surveys composed of questions in the language and style of the adult and the ethnic majority are unlikely to represent the full breadth of the population. The findings have been important for deriving general guidelines for forest management, which we briefly summarize below.

## 3. Review Results and Discussion

### 3.1. Land Ownership and Incentives for Forest Management Considering Recreational Interests

Among the landscapes that attract visitors seeking recreational opportunities, forests and woodlands offer a wide range of possibilities and are therefore especially popular [17,28,29]. Internationally, outdoor recreation is often concentrated in public forests because access to private property is limited [30,31], however, access to privately owned forests and woodlands is a legal right enjoyed by residents in many countries. Forests (both private and public) located near cities generally receive more visitors due to their proximity to dense populations and relative accessibility [32–36]. While city parks are publicly owned spaces that are managed with obvious regard for public access, private urban forests and woodlands are rarely managed for outdoor recreation [35,37]. In areas where public access to private lands is granted by law, people do not have to consider land ownership, but different owners can have different management priorities. More than half of the forested area in the Nordic region and Europe (excluding the Russian Federation) is privately owned, but typically allow access [32,34]. Here property boundaries can be subtly or inconspicuously marked such that only landowners and specialists are able to recognize them, and visitors may hence be largely unaware of the ownership of the forests they visit [31].

Private landowners and their associations have an understandable interest in maximizing their financial return from timber harvest. Because privately owned forests also provide publicly utilized products, a range of mechanisms often exist to generate incentives for landowners to manage their

properties with consideration of the public good [38] (Figure 2). Legislation, certifications, societal norms, and market incentives concerning forest management effectively set the context for which management decisions are acceptable regarding timber harvest rates, size of logging operations, and conservation of intact forest stands. Beyond meeting these minimum standards, landowners may make additional contributions or concessions that benefit the public interest. Yet the line between what constitutes a legal obligation and a voluntary contribution is often unclear. For example, Norwegian landowners operate within a principle of "freedom with responsibility", providing landowners with reduced bureaucratic oversight in exchange for an understanding that they will manage their properties according to certain concessions towards public use and enjoyment [39].

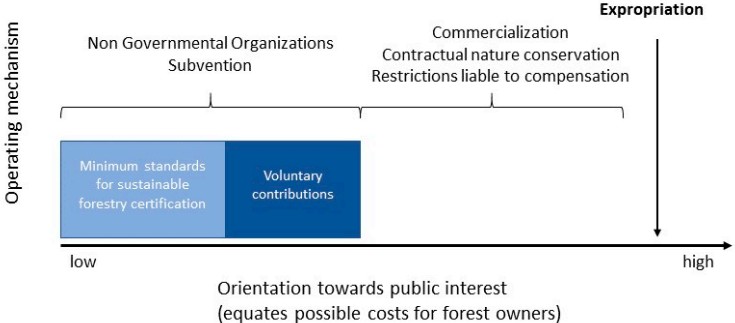

**Figure 2.** Forest management decisions oriented towards public interest (with respect to either ecological or social levels of management) represent increased contributions from, and costs to landowners. Operating mechanisms serving as regulatory or economic incentives change as forest management decisions become increasingly publicly oriented, with expropriation representing the most extreme cases, after Weber [38].

Whereas the forestry regulations that establish the minimum standards for acceptable management decisions are designed to simply avoid conflicts between landowners and public interests for recreation [40], landowner contributions beyond this minimum represent opportunities to add value to their properties. Enjoyable and rewarding recreational experiences on private lands could represent a market opportunity that might eventually develop a customer base which identifies with the forest products from a particular area and has a greater understanding of the sources of the products they consume [33]. Accordingly, efforts made to enhance recreation in areas that are protected from harvest can increase the value of the timber from portions of the property that are harvested. In cases where management decisions favoring recreation or other public interest result in considerable expense to landowners, either through lost harvest potential or an investment in infrastructure intended for public use, landowners might have a claim to government compensation in return for the public services that they provide [38] (Figure 2).

### 3.2. Forest Preferences and Forestry

People's visual preferences related to forests have been intensively studied in Fennoscandia since the 1970s, and to date we have identified 104 scientific peer-reviewed studies from Finland, Norway, and Sweden. The results from these surveys have direct influence on forest silviculture, first of all for general recommendations that are adopted by forest management [41] (Table 1). Several replicated surveys have shown that people's preferences are quite stable over time [42,43]. A common feature in the surveys is that people's preferences for a forest stand increased with increasing tree size and an advancing stage of stand development [44]. Many surveys indicated that the public tended to value irregular and/or multi-layered stands with a mixture of trees of different sizes. Semi-open stands that elicited a feeling of visual accessibility and the provision of a view were also very important. This suggests that people seem to prefer late succession unmanaged stands that are relatively open, and managed stands with large trees, including park-like forests in which the within-forest visibility

should be relatively good. On the other side, large clear-cuts and obvious traces from forest operations were unattractive. It is also interesting to note that trained foresters were more positive towards clear-cuts and traces from forestry operations than the general population, but these groups have also much in common regarding landscape appreciation [45]. Results from preference studies are suitable to integrate into practical recommendations focusing on forest elements and structures (Table 1).

**Table 1.** General considerations in forestry for people's preferences that are derived from forest preference research from Finland, Norway, and Sweden (at least three different surveys).

| General Considerations |
| :---: |
| Forest visitors view large, recent clearcuttings as negative elements. Seed trees or other retained trees usually make their impression somewhat better. |
| Forests with diverse topography and natural viewpoints are very much appreciated. |
| Grazing pasture and hayfields located in forests are regarded as positive elements. Evidence of historical uses provide a richer experience to many forest visitors. |
| Highly visible tracks from logging machinery operations give a negative impression. |
| In addition to factors such as openness, stand structure, and light conditions, people's preferences for tree species composition of a forest stand are influenced by what tree species the respondents are accustomed to. Elements of broadleaves in coniferous stands are appreciated by most visitors. |
| Many forest visitors prefer some level of visibility in forest stands, with view distances of approximately 30 to 40 m being most highly preferred. Forest structures that are too dense or too open forest are less preferred. |
| Most people oppose the use of herbicides and heavy soil scarification. |
| Most people prefer hiking on simple paths when visiting forests, despite that behavioral studies reveal that forest visitors to a large extent walk on walkways and forest roads. |
| Openings resulting from natural processes, such as bogs, heaths, and lakes are considered more attractive than openings resulting from clear-cuttings. |
| Professional foresters become significantly more enthusiastic than other people when exposed to photos of forest stands that have been treated in accordance with the syllabus from their forestry education. |
| Selective cuttings usually do not cause significant negative reactions among the general public. In general they are much more appreciated than clearcutting. |
| Stands containing snags and coarse woody debris are not appreciated by the general public, especially without information about the ecological importance of such elements. |
| Tending of young stands and different kind of thinning methods improve visibility and accessibility, and are generally accepted by most visitors if debris is removed after the thinning. |
| There is a positive correlation between tree height and perceived attractiveness. |
| Visitors prefer semi-open mature stands over dense, young stands. |
| Visitors tend to preferred multi-layered forest stands if visibility is maintained. |

There is, however, substantial variation among the surveys with regard to the methods employed and the external validity of their results, and both strong and weak points of the various designs have previously been discussed [44,46]. Nevertheless, preference studies have had an influence on regulations and standards for sustainable forestry in general and on urban forestry in particular [17,37,47].

*3.3. Forest Preferences and Biodiversity*

The general view that forest managers and forest owners had serious objections against the environmental movement in the 1970s has been partially supported by conflict investigations (e.g., [40]). Similarly, forest preferences among forest managers differed from those of other groups of people [45,48]. In Norway in the 1970s, foresters maintained that a sustainable forest management (by their narrative and definition) would in effect also take care of the public's moderate aesthetic demands, as well as ecological goals. However, new objectives in forestry related to sustainability and biodiversity

could lead to contradictions and conflicts with those who primarily take to the forests for aesthetic appreciation and outdoor recreation. A complicating factor is that the perception of attractiveness is very diverse, and depends on knowledge. On one side, perceived forest beauty and ecological sustainability can be conflicting concerns, but on the other side perceptions of beauty often rest on a specific ecological understanding of nature (see references [25,26,49]). This latter perspective often referred to as ecological aesthetics and is synthesized in a compilation of Aldo Leopold's writings (e.g., reference [26]). Ecological aesthetics separates the abstraction of people's experience in which the pursuit of pleasure is derived directly from viewing the landscape from that of (indirectly) knowing that the landscape is ecologically viable [25,50]. However, in practice it is not straightforward to distinguish the relationship between scenic preferred landscapes and non-visual environmental aesthetics based on an understanding of ecological sustainability [23]. For our framework, we take the position that preferred landscapes can lead people to form emotional attachments to the land and through this have larger engagement for sustainability on forested lands.

Conflicts were more obvious regarding large-scale even-aged silviculture [40]. However, conflicts between amenity and biodiversity could become more resolved in the future, as a main goal in forestry is to increase forest elements, forest structures, spatial arrangements, and processes important for biodiversity [41] (Table 2). The visual accessibility in urban woodlands will decrease as the proportion of protected forest increases, including processes such as self-regeneration, self-thinning, and gap and log formation. "Messy" forest structures such as dead wood, and processes including natural fire stands and post-fire successions are also less appreciated by people [20,24,26]. However, preferences were found to increase when respondents were given information about the ecological effects of the scenario [24]. People seem to appreciate the structural diversity and species diversity within forest stands, including large trees, multi-layered structures and rich herb layer up to a certain level [51,52]. However, too much structural diversity and complexity yielded lower preference scores, presumably because increased complexity will cause low coherence and readability of the forest environment. Moreover, studies of different natural scenes, such as garden landscapes, rivers, and brownfields restoration have demonstrated that higher ecological quality (rated by experts) tend to be more preferred than scenes with lower ecological quality [53,54]. Hence, studies suggest that when people are aware of the ecological quality of the scene, they are likely to display higher preference for high biodiversity settings, whereas high biodiversity scenes may receive lower preference ratings without this information [24]. The link between ecology and aesthetics also includes a part of education in the sense that people can perceive and learn about the natural environment including high levels of biodiversity.

**Table 2.** Structural components, spatial patterns and processes of importance for biodiversity in natural boreal mixed forests [41], and people's preferences for these components in boreal forests. Gundersen and Frivold [44] and Zachrisson et.al. [55] are used as the only references from studies before 2008, including a review of 53 scientific studies.

|  | Descriptions | Preferences | References |
|---|---|---|---|
| **Structural Elements Components/Elements** | Very old pine and spruce trees | Positive | [44,55] |
|  | Old broad-leaved trees, particularly *Populus tremula* and *Salix caprea* | Positive | [44,52,55] |
|  | Trees with abundant growth of epiphytic lichens | (Positive) | [44] Not well studied |
|  | Broken, stag-headed and leaning trees | Negative | [24] |
|  | Trees with holes and cavities | (Positive) | [44] Not well studied |
|  | Fire-scarred trees, snags and stumps | Negative | [20,24,44] |
|  | Downed logs (Large, Fresh) | Negative | [20,24,44,55] |
|  | Downed logs in various stage of decomposition | (Negative) | [20,24] (but strongly decomposed best liked) |
| **Spatial Patterns** | A developed understory of trees, saplings and shrubs | Negative | [52] |
|  | Mixed stands, with both conifers and broad-leaves | Positive | [44,55] |
|  | Uneven-aged stand structure | Positive | [44,55] |
|  | Multilayered tree canopies | Positive | [44,55] |
|  | Patchy distribution of trees | Positive | [44] |
| **Processes** | Post-fire succession | Negative | [20,24] |
|  | Succession with tree-species replacement (i.e., birch below spruce) | Unclear | [44] Not well studied |
|  | Self-thinning | Negative | [20,24,44] |
|  | Gap formation | Positive | [20,24] |
|  | Snag and log formation | Negative | [20,24] |
|  | Decomposition of coarse woody debris | (Negative) | [20,24] |

### 3.4. The Recreational Opportunity Spectrum (ROS)

Outdoor recreation comprises a wide range of activities, and not all landscapes are equally well suited for every activity. The reasons visitors have for visiting natural areas can in theory be as diverse as the visitors themselves [56,57]. The issue of outdoor recreation includes basic understanding about human perception of the environment, the role of place and landscape for identity, and the evolution of a modern commercialized society. Important traditions in planning for outdoor recreation are however largely based on empirical surveys of motivation, behaviour, attitudes, and expectations of forest visitors and stakeholders in a context of spatial conflict resolution and area management (e.g., reference [58]).

ROS is identified to be one of the most promising land use planning methods and includes a zoning approach to characterize the diversity of recreational activities and identify which environments visitors might seek for pursuing particular activities [59,60]. The combination of physical, social, and land-management aspects of an area determines the recreational opportunities that are present there, allowing areas to be classified into categories along a spectrum ranging from city parks to wilderness. Other monitoring and land assessment methods, including the Purism level [61,62] and the wilderness perception mapping concept [63], incorporate similar models for viewing landscapes across the continuum of increasing human influence. ROS has proven to be an instrumental approach that has commonly been adapted to a diversity of themes and places (e.g., [64]).

The ROS concept is based on an activity-opportunity definition of recreation, implying that users are seeking opportunities for activities, experiences, and benefits in particularly environments [57]. The central assumption for the method is that a visitor chooses to perform a certain activity in an appropriate environment to obtain a certain experience. A continuum of experiences in demand should match a continuum of available forest settings for the area in question. People who demand a wilderness experience should be provided with areas that offer solitude and few facilities, while people who prefer easily accessible areas with modern facilities and services should be offered their favourite types of places at the other end of the spectrum. The ROS formalizes this need for diversity by offering an assortment of recreational opportunities. These could include recreation opportunity

classes, commonly ranging from what many authors have labelled urban to primitive, organized diversity into land area units with different levels of physical alterations to the environment, different levels of remoteness, of size, of encounters with other users, and differences in the amount and type of management actions. The recreation settings are defined by:

- physical attributes, including topography, vegetation, different constructions, and impacts from harvesting
- social settings, including type of activities in demand, user density, crowding, and interactions between users
- managerial conditions, including management regulations and orders.

The number of recreation opportunity classes that comes out depends on the landscape qualities of the area, the recreational needs that are recognized, and the planner.

## 4. Management Implications

### 4.1. The Simple Four-Sector Framework

Our framework identifies four broad categories for characterizing the suitability of forests and woodlands for a range of recreational activities (Figures 3 and 4). The framework classifies areas within a conceptual space defined by using two axes: (1) the spectrum of human influence similar to those used in ROS classifications from urban/developed areas to wilderness areas, and (2) the level of economic contribution required by landowners to meet the recreational demands by visitors to their lands. We expect landowner contributions and the corresponding expenses to be commensurate with visitors' recreational preferences or expectations as these preferences increase from general to specialized.

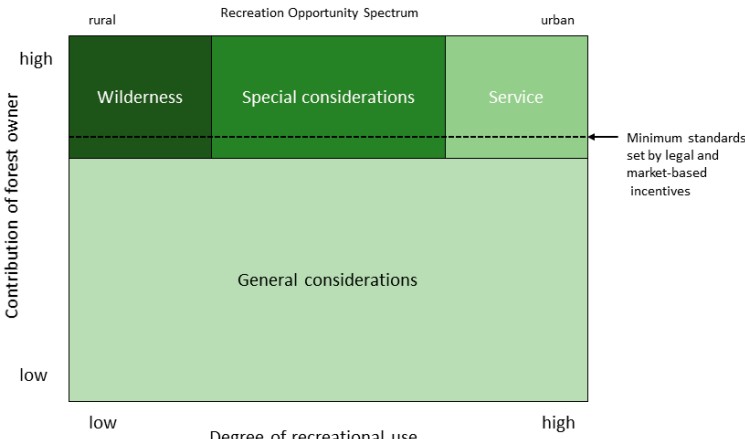

**Figure 3.** Conceptual model for simple four-part spatial zoning by combining values for recreation, biodiversity and timber resource utilization on privately owned forests. General consideration includes basic forestry adaptation to the sustainable use of forest resources defined by a forestry certification scheme. Wilderness, special consideration, and service areas include categories along the recreation opportunity spectrum (ROS) concept, providing recreational experiences and biodiversity conservation by combining environmental, societal, and management settings (modified after reference [17]).

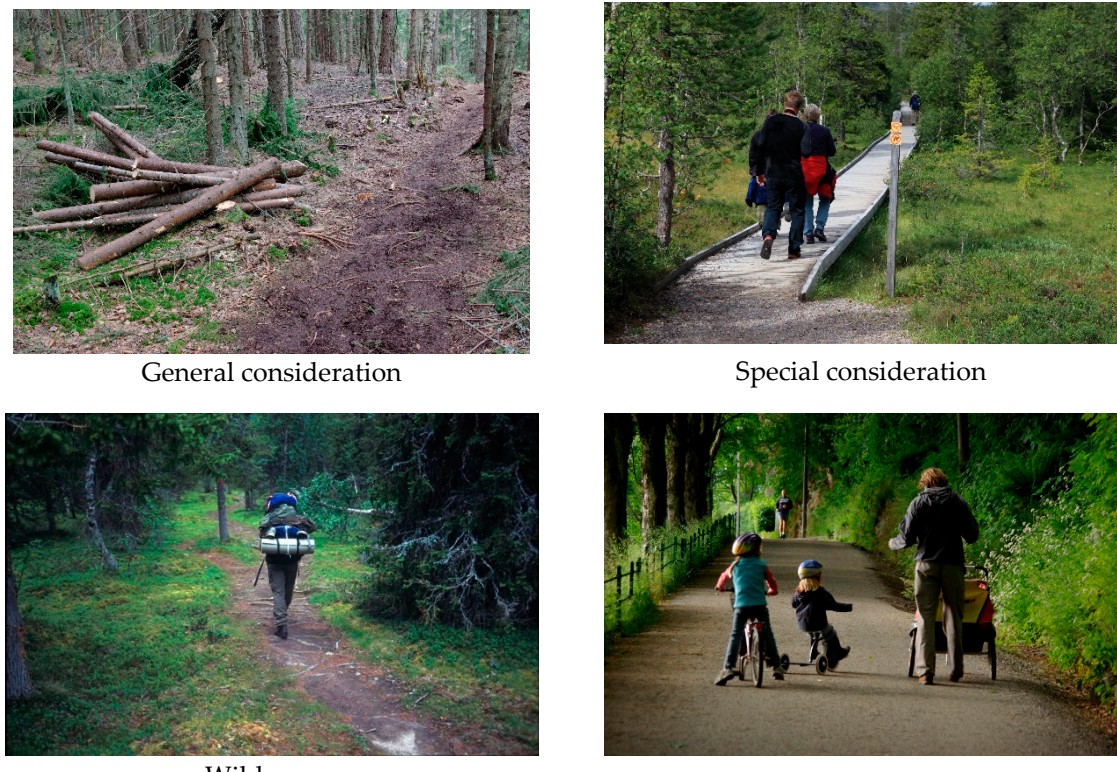

**Figure 4.** Four photographs depicting recreational infrastructure in the different zones of forests (Photos: Author).

The framework's first, and typically the largest, category comprises forested lands where general considerations for biodiversity conservation and access to recreational opportunities apply. Landowner contributions to enhancing recreational opportunities in these forests are generally modest, and the incentive to make such contributions is dictated by local or national laws and regulations concerning forestry practices. Additional incentives may come from a desire to meet conventional standards for environmental certifications by approved organizations such as the Forestry Stewardship Council and the Programme for the Endorsement of Forest Certification Schemes [65,66]. Ordinary forest land with minor importance for recreation and biodiversity can be placed in this category and comprises the largest proportion of forested lands [34,35]. Examples of contributions that landowners might make to enhance recreational opportunities include granting permission to establish recreation-oriented infrastructure on their property such as parking facilities, simple walking trails, and ski trails with modest signage, and making relatively minor adaptations to the harvest practices and silviculture in the relevant portions of their properties (see Table 1).

The second category describes areas with special considerations forest recreation and/or biodiversity (Figure 3). Special consideration regarding biodiversity includes silvicultural adaptations in or adjacent to sensitive areas such as riparian zones, rare forest types, agricultural landscapes, and critical wildlife habitats. Visitors' recreational demands and forest preferences in this category are such that landowners' contributions would—in most cases—exceed those expected by regulations or the forest products market incentives. Actions might involve an extended rotation age for trees, strict limitations on the prevalence and size of clear cuts, and tending or thinning to keep stands more open than the optimal density for wood production (Table 1), in line with aesthetic preferences. Typical examples of recreation-oriented facilities would include marked paths and information signs describing natural or cultural history in an aesthetically pleasing forest environment. Such management decisions and infrastructure investment typically imply significant reductions in the profits from timber sales [39], and are therefore easiest to implement in publicly owned forests or in forests where income

from tourism could compensate for reduced income from timber [67]. These special consideration areas could be easily accessible or otherwise particularly attractive forests with intermediate to high visitor intensity.

The third category consists of wilderness and nature reserves: areas where management decisions are intended to protect biodiversity and allow visitors pursuing recreation to experience the character of natural ecosystems while not necessarily facilitating high volumes of visitors (Table 2). Forests in this category would not provide landowners with income from timber harvest, and other economic incentives would need to be enlisted. In most countries, this category generally represents a small proportion (between 1% and 5% of forested lands) of public and privately owned forests [68]. The low intensity of recreational use that characterizes these forests should not, however, be confused with marginal public interest in the recreational opportunities available in wilderness forests [69]. A large proportion of people in Finland perceive wilderness as roadless, uninhabited areas covered with virgin forests and natural mires, and believe that the areas must be quite large and silent. Another important criteria is that such areas must lie rather far away from roads and inhabited areas [70]. However, people's perception of what constitutes wilderness depends on context, is dynamic over time, and for many people also includes areas of former selective cuttings and elements of old cultural landscapes [70]. In the 1980s wilderness areas in Fennoscandia were defined as very large remote areas, but today wilderness areas are often mentioned in the literature as rather small patches (e.g., 100 ha) of untouched forest landscapes that are close to urban areas [17]. People in Finland have been rather keen visitors to wilderness areas: about fifty percent of the respondents in a nationally representative survey had visited large wilderness areas during their lifetimes [70]. Additional, almost all the respondents (96%) expressed in this study that they support wilderness conservation. Natural forests are the most important features of the non-mountainous wilderness areas in Fennoscandia—generating images of vast, pristine areas without the intrusion of roads [55,68]. It is also important that pristine and wilderness forests are not limited to areas located far from population centres. Some small wilderness areas or "wild corners" in urban woodlands should be maintained or developed close to where people live, both for the sake of biodiversity and for the residents' experience of nature and wilderness. If wilderness areas are located where they receive large numbers of visitors, these landscapes may require recreational infrastructure and directed management measurements to channelize visitors such that the area retains its untouched character [17].

The fourth category consists of service areas, where the intensity of recreational use is highest, and where forests receive the greatest numbers of visitors. These forests require park-like management, and may include improved signage; designation of special activities such as areas for swimming; and infrastructure such as sport facilities, visitor centres, and hotels. As with the two previous categories, even-aged silvicultural and harvest methods are generally incompatible with the visitors' demands and preferences for aesthetics and specific recreational activities. Yet in service category forests areas, landowner contributions and lost revenue from timber harvest can often be more easily compensated for by the sale of activities and the use of infrastructure (e.g., parking fees and access to activities) or funds from the authorities (e.g., to stimulate outdoor recreation). "Messy" forest structures that are less appreciated such as dead wood, dense young-succession forests, and snags should either be removed if they pose a hazard to safety, or be a part of a public information strategy for enhancing the appreciation for biodiversity. Areas in this category are usually not very large and can be located in areas that are both close to high population centres as well as areas that are far more remote, such as in the entrance areas of national parks.

### 4.2. Management Implication and European Relevance

The three considerations categories along the ROS spectrum (wilderness, special considerations, and service areas) ask for a spatial segregation of forest functionality with variable influence by humans. These three categories cover a minor part of the landscape in most cases, often connected to entrance areas for tourism and recreation, fringe areas around towns and cities, and other important areas

including human presence and influence. In contrast, areas with general consideration typically cover large tracts of forest land and involve the integration of timber production, conservation of biodiversity, and opportunities for outdoor recreation in a manner that is weighted more uniformly across the landscape. Thus, the conceptual model involves both segregative and integrative instruments [71] that cover areas with special interest values and areas with general considerations in a multi-purpose forestry. We hypothesize that these various instruments complement each other in a forest management situation, i.e., maintaining the different aspect of conservation and recreation across different spatial scales and degrees of silvicultural use intensity.

Because public forest preferences have been an important research topic in the boreal forest system, along with the development of a theoretical framework for management of recreation along a spectrum of users (e.g., ROS), it is interesting to discuss the transfer value to other European forest ecosystems. The spectrum of recreational use along the naturalness-development continuum (from wilderness to service areas) [17] seems to correspond quite well with people's preferences for forest environments at a European scale [55,72,73]. It is likely that on a coarse scale, generalizations can also be made about the preferences of forests as venues for recreation across Europe [55], but it is important to consider the site conditions, history of use, management, and cultural differences in different regions [71,74,75]. Of five archetypical recreation user groups in the European Union (The convenience recreationist, the day tripper, the education recreationist, the nature trekker, and the spiritual recreationist), the nature trekker has a higher likelihood to thrive in a boreal forest due to factors such as remoteness that may be more challenging for the other groups to deal with [76]. Additionally, the nature trekker has a high potential in mountainous areas throughout Europe (e.g., the Highlands of Scotland, the Alps, the Pyrenees, and the Carpathians), with similar forest qualities to the boreal forest [77]. We therefore hypothesize that our conceptual model is context-specific and primarily adapted to Fennoscandian boreal forests, but that it may be transferable to other mountainous areas in Europe on a smaller spatial scale.

### 4.3. Concluding Remarks

The primary aim of the framework presented here is to inspire landowners to view their forested properties as areas that can offer a range of recreational opportunities to the general public, with varying degrees of human influence or infrastructure development, and that this can be organized along an urban-wilderness gradient that is important for recreation and biodiversity. This simple conceptual framework also presents a tool that land managers can use for planning and zoning decisions. Because the amount of human influence is explicit in the various categories, land managers can use the framework to establish targets for the proportion or total area that might be classified into each category. Such forest-recreation category designations might also be applied to the land appraisal process, identifying areas where landowners might be eligible for compensation for the contributions they have made or could make to enhance the public services available on their lands. We present this framework here in its most simple form, and acknowledge that including additional aspects of recreational land use will help refine and improve its applicability. Specifically, there is a need for translating the findings from disciplinary and interdisciplinary research to practical management approaches, and to monitor the effects where such regimes have been implemented. Future research should investigate the consequences of such models on the production of timber, recreational opportunities, and conservation of biodiversity.

**Author Contributions:** V.G. conceptualized the model and do the original draft preparation. B.K. and K.M.M. review and edit the draft paper and improve the review methodology and the concept. All authors commented on and approved further drafts.

**Funding:** The Research Council of Norway funded this research within the project "SIS-URBAN".

**Acknowledgments:** We want to thank two anonymous reviewers for their constructive criticism and suggestions how to clarify our arguments.

**Conflicts of Interest:** The authors declare no conflict of interest.

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
