# Peer review of "Seeing the Forest for the Trees: A Review-Based Framework for Better Harmonization of Timber Production, Biodiversity, and Recreation in Boreal Urban Forests"

_urbansci, doi:10.3390/urbansci3040113_

Round 1

Reviewer 1 Report

This paper addresses an important topic: balancing the multiple benefits provided to humans provided by boreal urban forests. The framework developed by the authors could prove of assistance to forest managers, but I find major room for improvements both in the conceptual development of the framework and the general presentation of the manuscript that require heavy revisions before it is suitable for publication.

Specifically:

The description of the methodology for the literature review needs to be fleshed out-what were some of the search terms? Common journals and authors? How was grey literature addressed? More detail is needed on the nested analysis than just one quick sentence. The justification for focus on Fennoscandian boreal systems runs quite long Lines 118-119: on-site, mailed, and web-based are not sampling frames Much of the text from the caption for figure one would be better suited to appear in the manuscript itself Lines 149-150: if the focus is on Fennoscandia why include Europe and Russia? Lines 175-179: I think this is quite a stretch. People will seek out forest products from areas where they recreate? Any citations to back this up? Lines 186-187: here it is more than 60 studies, where before it was 104? Table one is not very reader friendly, needs major reformatting Same with table two Figure three: according to the text (in some places) special considerations should have lower contributions from forest owners but they are depicted as having the same level here Figure three: scale would be another important variable to incorporate if possible Lines 343-346: I’m note sure if the survey cited is good evidence that people in Finland are rather keen visitors of wilderness areas. Rather keen compared to what? Are they really keen if they have been to one once in their lives? Lines 357-359: special considerations are presented above as changing production practices but not eliminating commercial harvest, here they are-need to be consistent Section 4.4: I’m not sure what saying that the model has been used in one location adds anything without some sort of analysis-how did it actually work? Did managers and recreationists like it? Etc. The conclusion is quite brief and doesn’t offer much guidance as to how the model can actually be used for management In general, the idea that managers would have different management types on their property based upon physical characteristics and demands is important, but dismissed early and not returned to

Author Response

Response to reviewer 1

This paper addresses an important topic: balancing the multiple benefits provided to humans provided by boreal urban forests. The framework developed by the authors could prove of assistance to forest managers, but I find major room for improvements both in the conceptual development of the framework and the general presentation of the manuscript that require heavy revisions before it is suitable for publication.

Specifically:

The description of the methodology for the literature review needs to be fleshed out-what were some of the search terms? Common journals and authors? How was grey literature addressed? More detail is needed on the nested analysis than just one quick sentence.

Authors: We have rewritten this chapter completely, described our procedure more in detail and inserted: “Reviews of the interface between timber production, recreation, and conservation have previously been published [8-13]. Consequently, we did not perform a comprehensive review in this paper, but merely focused on influential reviews and seminal papers as a starting point for a targeted search in a restricted geographical area. However, for the quite narrow topic people`s forest preferences we did a comprehensive search with the aim to include all relevant scientific papers from Fennoscandia. For this we used four international databases (Web of Science, Google Scholar, Oria, and Scopus) with a diversity of terms : preference, perception, attitude, like, dislike, visual, scenic, appreciation, aesthetic, and expectation with the word boreal, forest, wood, park, and woodland in combination with Norway, Sweden, Finland, and Nordic. We included all peer-reviewed papers that addressed people’s visual preferences for forest since the topic first appeared in 1972. We evaluated 152 papers, resulting in 104 papers for this review (Figure 1). Some of the papers included two or more surveys; some papers were based on the same dataset; and some of the papers focused on other landscape components (urban parks, rivers, agricultural lands) but included important results about visual forest attributes. The 48 papers that were excluded from the review provided important baseline information about mechanisms for forest visual appreciation, but did not directly investigate peoples visual preferences. We focused on the boreal forest in Fennoscandia (Finland, Norway and Sweden) as integrating these values has been an important issue over the last fifty years [17]. Clear-cuttings and constructions of forest roads have caused large conflicts between forestry and recreationists, and there was a need for knowledge about people’s preferences to solve these practical problems in forestry [28]. The survey methodology has varied substantially, using different kinds of target populations (national, regional and local), sampling modes (on-site, mailed, web-based), and stimuli (photos, verbal, on-site in the forest environment).”

Regarding nested analysis we deleted this sentence:

“To ensure we obtained the most relevant literature we performed a nesting analysis in which we reviewed the literature forward and backward in time from the publication date of selected papers.”

The justification for focus on Fennoscandian boreal systems runs quite long

Authors: Yes, we agree on that, but decide to keep the text. The argument for this is that we feel it is important to describe similarities and differences between the Fennoscandian culture/ecosystem and elsewhere in Europe, for the understanding of relevance and possibilities for transforming results. We decide to include this in a new chapter 2.1.

Lines 118-119: on-site, mailed, and web-based are not sampling frames

Authors: Very good observed. It should be “…sampling modes…” and we have changed this.

Much of the text from the caption for figure one would be better suited to appear in the manuscript itself

Authors: Agree. We incorporated this text in the chapter “2.2. Reviewed literature”: “Some of the papers included two or more surveys; some papers were based on the same dataset; and some of the papers focused on other landscape components (urban parks, rivers, agricultural lands) but included important results about visual forest attributes.”

Lines 149-150: if the focus is on Fennoscandia why include Europe and Russia?

Authors: We find it here relevant to put the Fennoscandian situation in a larger European context, because the private ownership is our basic approach for our model, and in this sense could be of relevance for other places in Europe.

Lines 175-179: I think this is quite a stretch. People will seek out forest products from areas where they recreate? Any citations to back this up?

Authors: We added a reference to underline this sentence, Rydberg & Falck 2000 discuss this. It is as simple that good forest experiences enhance the legitimacy to harvest from the forest.

Lines 186-187: here it is more than 60 studies, where before it was 104?

Authors: Thank you. Yes, this is an old number for our first search for research on visual forest preferences in Fennoscandia. We extend the terms that we searched for and included in all 104 papers. This is the wrong number and has been changed.

Table one is not very reader friendly, needs major reformatting

Authors: Agree. We have rewritten and reformatted table 1.

Same with table two Figure three: according to the text (in some places) special considerations should have lower contributions from forest owners but they are depicted as having the same level here

Authors: Contribution from forest owners on special consideration is also significant, especially regarding biodiversity. The text lacks better specification of this and we added this sentence:

“Special consideration regarding biodiversity includes silvicultural adaptations in or adjacent to sensitive areas such as riparian zones, rare forest types, agricultural landscapes, and critical wildlife habitat. ”

Figure three: scale would be another important variable to incorporate if possible Lines 343-346: I’m note sure if the survey cited is good evidence that people in Finland are rather keen visitors of wilderness areas. Rather keen compared to what? Are they really keen if they have been to one once in their lives?

Authors: Hallikainen (1998) is one of the most comprehensive and elaborated study of socio-cultural aspect of wilderness in the Nordic, so we add some more sentences to underline some important point regarding this. Yes, scale is important and it is a good idea to elaborate more about this. We added these sentences:

“A large proportion of people in Finland perceive wilderness as roadless, uninhabited areas covered with virgin forests and natural mires, and the areas must be quite large and silent. Another important criteria is that such areas must lie rather far away from roads and inhabited areas [69]. However, people’s perception of what constitutes wilderness depends on context, is dynamic over time, and include for many people also areas of former selective cuttings and elements of old cultural landscapes [69]. In the 1980s wilderness areas in Fennoscandia were defined as very large remote areas, but today wilderness areas are often mention in the literature as rather small patches (e.g. 100 ha) of untouched forest landscapes close by for example urban areas [17].”

Regarding the discussion about whether Finnish people are rather keen visitor or not, we assume that this is left to the reader, as more than 50 % of the visitor have visited wilderness areas. We added this sentence:

“Additional, almost all the respondents (96 %) expressed in this study that they support wilderness conservation.”

Lines 357-359: special considerations are presented above as changing production practices but not eliminating commercial harvest, here they are-need to be consistent

Authors: Good observation. Commercial forests in Fennoscandia are totally dominated by even-aged forestry, so in this context we formulated that sentence. We have changed the sentence:

“…even-aged silvicultural and harvest methods are generally incompatible with the visitors’ demands and preferences for aesthetics and specific recreational activities.”

Section 4.4: I’m not sure what saying that the model has been used in one location adds anything without some sort of analysis-how did it actually work? Did managers and recreationists like it? Etc. The conclusion is quite brief and doesn’t offer much guidance as to how the model can actually be used for management In general, the idea that managers would have different management types on their property based upon physical characteristics and demands is important, but dismissed early and not returned to

Authors: Agree, we deleted chapter 4.4.

Authors: Regarding your point about conclusion, we have added some sentences:

“The primary aim of the framework presented here is to inspire landowners to view their forested properties as areas that can offer a range of recreational opportunities to the general public, with varying degrees of human influence or infrastructure development, and that this can be organized along an urban-wilderness gradient important for recreation and biodiversity. This simple conceptual framework also presents a tool that land managers can use for planning and zoning decisions. Because the amount of human influence is explicit in the categories, land managers can use the framework to establish targets for the proportion or total area that might be classified into each category. Such forest-recreation category designations might also be applied to the land appraisal process, identifying areas where landowners might be eligible for compensation for the contributions they have made or could make to enhance the public services available on their lands. We present this framework here in its most simple form, and acknowledge that including additional aspects of recreational land use will help refine and improve its applicability. Specifically, there is a need for translating the findings from disciplinary and interdisciplinary research to practical management approaches, and to monitor the effects where such regimes have been implemented. Future research should investigate the consequences of such models on the production of timber, recreational opportunities, and conservation of biodiversity.”

Reviewer 2 Report

I believe this paper is publishable, but in its current form, there are some important improvements which are needed, see my comments below:

Title: make it explicit that the manuscript is a review.

Abstract: it could be improved for readability, insert the word “review”

Keywords: insert Fennoscandia;  it is leisure time

Page 2 line 66: insert citation

Reviewed literature and methodology: need major rewriting for sense and flow

Table 2: Structural….?

Page 8 line 291: discuss your review results

Discussion and conclusion: Some study implications and limitations are not clearly presented.

Page 11: discuss climate change implications

References: see Mery et al. (2010) Forests and Society – Responding to Global Drivers of Change. IUFRO World Series Volume 25. Vienna. 509 p.

Brack 2019 Forests and Climate Change https://www.un.org/esa/forests/wp-content/uploads/2019/03/UNFF14-BkgdStudy-SDG13-March2019.pdf

Author Response

Response to reviewer 2

I believe this paper is publishable, but in its current form, there are some important improvements which are needed, see my comments below:

Title: make it explicit that the manuscript is a review.

Authors: Good, we added “review-based” and here is the title: “Seeing the forest for the trees: A review-based framework for better harmonization of timber production, biodiversity and recreation in boreal urban forests”

Abstract: it could be improved for readability, insert the word “review”

Authors: We have rewritten some sentences in the abstract to increase the readability.

Keywords: insert Fennoscandia; it is leisure time

Authors: Done, and we have organized the keywords in alphabetical order

Page 2 line 66: insert citation

Authors: Good. Done

Reviewed literature and methodology: need major rewriting for sense and flow

Authors: Agree. We have completely rewritten this chapter focusing on a better description of our methodology in general and literature search in special. To increase the readability we have split the methodology chapter in the two sections; “2.1. Delimitation of the study” and “2.2. Reviewed literature”. We think the first chapter is important to describe the frame and relevance for our paper.

Table 2: Structural….?

Authors: Good. We have changes to “Structural elements”

Page 8 line 291: discuss your review results

Authors: Yes, this is a good point. Most of the discussion of the review is in chapter 3. Review results, so we changed this heading to “3. Review results and discussion”. The presentation of the model is thus more like a conclusion, and we changed the heading of this chapter to: 4. Management implications. We suggest this will better represent the content of the different chapters.

Discussion and conclusion: Some study implications and limitations are not clearly presented.

Authors: Good. We have added some sentences here and there to clarify the content in this chapter. We also decide to delete chapter 4.4. See track changes for changes that are done.

Page 11: discuss climate change implications

References: see Mery et al. (2010) Forests and Society – Responding to Global Drivers of Change. IUFRO World Series Volume 25. Vienna. 509 p.

Brack 2019 Forests and Climate Change https://www.un.org/esa/forests/wp-content/uploads/2019/03/UNFF14-BkgdStudy-SDG13-March2019.pdf

Authors: Yes, we agree that climate change is a very important factor regarding management of urban woodlands in Fennoscandia, however, we feel that such a large theme needs to be presented more throughout the paper, and as an aim from the beginning. Therefore, we decided to not include climate change as a driver and factor in our paper. But if you have ideas on how this could be included in a way that wouldn`t need to much space (the paper is already at 8447 words), and at the same time add relevant knowledge for the implementation of the model or for the implication of the model, please let us know.

Round 2

Reviewer 1 Report

this is a major improvement and I think the article is acceptable for publication

Reviewer 2 Report

The authors have addressed referees’ comments with detailed replies to each of the issues raised.   The manuscript has been improved and can be published.

This manuscript is a resubmission of an earlier submission. The following is a list of the peer review reports and author responses from that submission.

Round 1

Reviewer 1 Report

The authors have provided a nice overview of the various human uses and preferences for management of boreal forests in Fennoscandia. This review would be informative for readers interested in understanding patterns of land ownership and how that relates to provisioning of recreation opportunities, and the characteristics generally preferred in such forests by respondents in previous published surveys.

The authors state in the Introduction that a spatial zoning approach is needed to balance the competing demands of timber production, biodiversity conservation, and access to outdoor recreation, and prepare readers to anticipate that they will be proposing a conceptual framework for characterizing the recreational suitability of forests and woodlands which will assist in balancing these demands. The paper falls short of this lofty goal, as the only 'framework' provided is a single simplistic diagram. Far more development of the concepts underlying this diagram would be needed to begin helping balance these competing demands. The ROS uses a large number of factors to characterize sites according to several complex spectrums, and the presentation of a single diagram does not adequately convey this sophistication.

The authors indicate in the Conclusion that "the primary aim of the framework presented here is to inspire landowners to view their properties as areas that can offer a range of recreational opportunities to the general public". I don't believe this brief summary of previous research and the one diagram describing the 'framework' would motivate landowners to see their properties in a new light.

Overall, I question if this manuscript is suitable for publication as a journal article. It seems more appropriate as a book chapter, and portions of it could also be suitable for some popular press (magazines, newsletters, etc.).

Reviewer 2 Report

Having read the second version of the manuscript leaves me somewhat perplexed. I recognize that the authors have made some effort to extend their manuscript to address some of the critical points. A short paragraph on management implications was added (lines 417-430), but it remains highly hypothetical.

I still miss the promised "management tool" (now line 454). I am not saying the presented conceptual framework cannot be useful for forest management decisions - this hypothetical usefulness is just not demonstrated. I am afraid I have to state that the authors have not sufficiently addressed the points I raised with the manuscript previously.

The litarature review is broad and most probably supports the framework in (what is now) figure 3. The manuscript is a good review paper and provides a classification of reviewed results. But as I commented on the earilier version, to establish a conceptual framework would need some empirical foundation or at least present ideas how this could be done.